# Physical Inactivity Amplifies the Link Between Anxiety, Depression, and Neck Pain in Breast Cancer Survivors

**DOI:** 10.3390/biomedicines13051089

**Published:** 2025-04-30

**Authors:** Guilherme H. D. Grande, Paloma Ferrero-Hernández, Leonardo R. de Oliveira, Vitória M. Danno B. Ramos, Mariana R. Palma, Rubens Vidal, Anna A. P. M. Oishi, Gerson Ferrari, Diego G. D. Christofaro

**Affiliations:** 1School of Technology and Sciences, São Paulo State University (UNESP), Presidente Prudente 19060-900, Brazil; marianaromanholi@hotmail.com (M.R.P.); rubensvcvidal@hotmail.com (R.V.); anna.oishi@unesp.br (A.A.P.M.O.); diego.christofaro@unesp.br (D.G.D.C.); 2Faculty of Medicine, University of Western São Paulo (UNOESTE), Presidente Prudente 19050-920, Brazilvitoriaramos36@gmail.com (V.M.D.B.R.); 3Vicerrectoría de Investigación e Innovación, Universidad Arturo Prat, Iquique 1110939, Chile; paloma.ferrero@gmail.com; 4Faculty of Health Sciences, Universidad Autónoma de Chile, Providencia 7500912, Chile; 5Escuela de Ciencias de la Actividad Física, el Deporte y la Salud, Universidad de Santiago de Chile (USACH), Santiago 7500618, Chile

**Keywords:** breast cancer, low back pain, neck pain, physical activity, anxiety, depression

## Abstract

**Background and Objectives:** Breast cancer can cause significant changes in both physical and mental health, leading to conditions such as low back pain, neck pain (musculoskeletal pain), anxiety, and depression. Women with this diagnosis tend to subsequently decrease their levels of physical activity, which can contribute to the emergence of musculoskeletal dysfunction. With this in mind, the aim of the present study was to analyze the relationship between physical activity levels (active vs. inactive), neck and low back pain in women breast cancer survivors in association with anxiety and depression. **Methods:** This cross-sectional study was conducted with 128 women breast cancer survivors. The prevalence of neck pain stratified by anxiety and depression status was higher in women physically inactive than in active women; however, there was no statistically significant difference. **Results:** Symptoms of anxiety and depression were associated with neck pain only in physically inactive women. No associations were observed between anxiety and depression and low back pain regardless of physical activity levels. **Conclusions:** This study demonstrates significant results for the association between physical activity level, anxiety and depression, and neck pain in women breast cancer survivors. However, the same association was not found for low back pain.

## 1. Introduction

Breast cancer (BC) is the most common malignant neoplasm among women worldwide, with over 2 million new cases, and more than 600,000 deaths from this cancer, in 2024 [1,2]. It is predicted that the worldwide incidence of female BC will reach approximately 3.2 million new cases per year by 2050 [3]. The global breast cancer mortality rate has increased by 0.23% per year, with statistically significant rises observed in individuals under 50 and over 70 years of age [4]. The incidence and mortality rates of BC have increased over the past three decades due to shifts in risk factor profiles, improvements in cancer registration systems, and advancements in cancer detection [5].

It is also observed that although the prevalence of this neoplasm is higher in developed countries, the highest mortality rates are observed in less developed regions [6]. Among the numerous risk factors for this disease, variables such as sex, aging, the female hormone (estrogen), family history, genetic mutations, and an unhealthy lifestyle can increase the possibility of developing BC [7,8,9].

The diagnosis of BC and the possible treatments used can have a large impact on quality of life, which may lead to the development of systemic side effects, perceived shoulder pain, and anxiety and/or depression, due to the stigma related to the pathology and the impacts of the therapy on body appearance and female identity [10,11,12,13,14].

Anxiety is an emotional state defined by tension, persistent worry, and physiological responses, such as elevated blood pressure. While anxiety and fear are distinct concepts, they are frequently used interchangeably. Anxiety is regarded as a future-focused and prolonged reaction to a vague or nonspecific threat [15,16,17]. Depression is described as a mood disorder that causes a persistent feeling of sadness and loss of interest, and which controls patients’ attitudes by altering their perception of themselves, so that they start to view situations in their life as major problems. This disorder presents characteristics that may indicate a serious pathology or simply another manifestation of the condition when faced with a real-life situation [18,19,20,21].

The condition of BC and the development of associated psychological symptoms tend to lead to decreases in the levels of physical activity in affected individuals, which can contribute to the emergence of musculoskeletal dysfunctions, including neck pain and low back pain [22].

Women more frequently report neck pain than men [23]. The causes of neck pain vary widely, with the main causes being poor ergonomics at work, when sitting, and maintaining neck posture in a non-physiological position for long periods of time [24]. There is, however, greater evidence for some risk factors, such as a lack of physical activity, duration of daily computer use, perceived stress, and being female [25]. As there is a tendency for neck pain to become a chronic problem, it is important to identify risk factors to enable its prevention and early diagnosis [26].

The biopsychosocial model postulates low back pain as a dynamic interaction between social, psychological, and biological factors that can predispose to and result from injury [27]. Low back pain is the leading cause of disability and loss of productivity worldwide, with a lifetime prevalence of up to 84% for the adult population [28]. This condition can be classified as mechanical, radicular (neuropathic), or due to increased sensitivity of the central nervous system, and the distinctions affect the treatment decisions [29].

Low back and neck pain are multifactorial diseases and a major problem in modern society, as well as being the leading and fourth leading causes of years lived with disability (YLDs) worldwide, respectively. Moreover, the prevalence of neck pain (NP) is exceeded only by depressive disorders and other musculoskeletal conditions [30].

It is widely recognized that physical activity contributes to improving symptoms of anxiety and depression and reducing musculoskeletal pain [31]. Ribeiro et al. [13], in a study involving BC women survivors, observed that the practice of physical activity during leisure time and commuting physical activity were inversely related to depressive symptoms. Mirandola et al. [32] observed that physical activity through exercise favored greater upper limb mobility in BC survivors, which could reduce the chances of musculoskeletal pain. The gap in our knowledge about the associations of low back and neck pain with anxiety and depression in women who have recovered from BC demonstrates a clear need for a study on the association of both conditions, since studies in this context are still scarce. The current work differs from previous studies, as it aims to understand if there is an association between the aforementioned symptoms and the decrease in the practice of physical activities. In this way, the results can help to identify the best therapeutic approach to promote biopsychosocial well-being. Therefore, the aim of the current study is to evaluate the association between anxiety and depression with low back and neck pain related to women BC survivors according to physical activity levels.

## 2. Materials and Methods

### 2.1. Study Design and Sample Selection Process

This is a cross-sectional study, approved by the Research Ethics Committee of FCT/UNESP (CAAE: 54169416.6.0000.5402). All volunteers who agreed to participate in the study signed an informed consent form. To calculate the sample, a correlation of r = 0.30 was considered between anxiety symptoms and low back pain based on the study by Hung et al., with an alpha error of 5% and a sample power of 80%. Anticipating possible losses, an additional 20% was added to the sample size, requiring a minimum of 101 participants [33].

### 2.2. Selection Criteria

The study was conducted with 128 female breast cancer survivors. Participant recruitment was conducted through institutions that support breast cancer and referrals from mastologists in the city. Data collection was carried out through face-to-face interviews conducted by previously trained researchers.

The selected inclusion criteria were women who had survived breast cancer and being over 18 years of age. The exclusion factors were having a pacemaker or metal plate, due to bioelectrical impedance, or any type of mental disorder that would prevent the woman from answering the questionnaires.

### 2.3. Data Collection

Personal information, such as name, age, marital status, and education, and clinical data, such as date of diagnosis and types of surgical treatment and alternative intervention were collected.

### 2.4. Physical Activity

The degree of physical activity was assessed using the Baecke questionnaire [34] which presents good reliability against gold standard methods like double-labeled water [30] and was ratified for the study population [35]. This evaluation method contains 16 questions divided into three areas of habitual physical activity, collected in relation to the year prior to the study [36]. In Section 1 of the questionnaire, the occupational score of physical activity is evaluated through eight questions that cover aspects of exercises performed during domestic services or work, such as amounts of weight carried, among others, checking for the presence of fatigue [36]. The second section is related to sports practices during leisure-time or physical exercises. It consists of four questions that classify the intensity (light, moderate or intense), frequency, and time attributed to the activities performed [34]. The composition of the points also considers comparisons in relation to leisure or work carried out by people of the same age and level of activity, while the final question considers the performance of exercises outside the leisure period. In Section 3, the degree of physical activity in leisure time and during transportation is measured by means of four questions about the frequency in which transportation is employed during these practices. This part considers the time spent cycling, watching TV, walking, and also the time it takes to shop or go to work. This tool provides an unfounded score through a specific equation for each sphere evaluated and the sum of the three points of the sphere establishes the total practice of physical activity, which can vary from 0 to 15 points [36]. As the instrument developed by Baecke et al. [34] does not provide specific cut-off points to classify physical activity levels, quartiles were used in the analysis. Women in the highest quartile (quartile 4) were classified as active (≥8.92 points according to Baecke score), while those in the lower quartiles were classified as inactive (<8.91 points according to Baecke score).

### 2.5. Socioeconomic Status

The Brazilian Economic Classification Criteria (ABEP) [37] was used to classify economic status. This tool performs an evaluation according to educational level, the number and type of rooms in the home, and the consumer goods in the household, in working order, regardless of how the goods were acquired. The instrument generates a score, which indicates the socioeconomic classification, stratified from the highest to the lowest classes (A, B1, B2, C1, C2, D, E). Based on these criteria, the participants in the current study were separated into classes: high (A and B1), medium (B2, C1 and C2), and low (D and E) [36].

### 2.6. Depressive Symptoms

The Hospital Anxiety and Depression Scale (HADS) was used to assess the presence of symptoms of anxiety and depression [38]. The main advantage of the HADS is its validity for use in clinical populations, as it was specifically designed to exclude items that could be generated due to physical conditions rather than psychological states [39]. The HADS questionnaire contains 14 multiple-choice questions, which separately assess symptoms of anxiety (seven questions) and depression (seven questions). Each item receives a score from 0 to 3, with a higher score representing a higher occurrence of anxiety and depression symptoms. The range for each outcome (i.e., anxiety and depression) is from 0 to 21 [40,41]. A score greater than or equal to nine indicates a potential case of anxiety and/or depression [42]. In the present study, the individual item scores were analyzed rather than solely considering the cut-off point for each symptom.

### 2.7. Musculoskeletal Pain

NP and LBP were assessed using the Nordic Musculoskeletal Questionnaire, designed to report musculoskeletal symptoms [43]. The questionnaire is divided into two parts: the first part is a general survey where the participant indicates the body part(s) in which they experience pain. The second part is a specific questionnaire where the participant answers questions about the duration of symptoms throughout their life, in the previous 12 months, and in the previous 7 days. In addition to symptom duration, the questionnaire also assesses the impact of pain on work and leisure activities, the presence of medical leave, and symptom duration. In this instrument, there are two possible answers: yes or no. In the present study, responses reporting cervical and/or lumbar pain in the last seven days were considered.

### 2.8. Statistical Analysis

The sample characterization variables are presented as mean and standard deviation for continuous variables and frequency for categorical analyses. The independent t-test was used to compare continuous variables, and the chi-square test was used to compare categorical variables. The association between anxiety and depression with neck and low back pain according to the physical activity levels of BC survivors was performed by Binary Logistic Regression in the unadjusted model and in the adjusted model (age, marital status, type of surgery, and socioeconomic classification). The statistical significance was set at 5% and the confidence interval was 95%.

## 3. Results

Table 1 presents the variables used to characterize the sample according to the levels of physical activity. The mean age of the inactive women was 59.06 (±10.03) and of the active women was 55.88 (±8.58). Most patients were married and had undergone a quadrantectomy. The prevalence of neck pain was 34.6%. Low back pain was reported by 56.2% of the study participants. No statistically significant differences were observed in the sample characterization variables when compared to physically active and inactive BC survivors.

In Figure 1, it can be seen that the prevalence of neck pain was higher in physically inactive patients than in physically active patients; however, there was no statistically significant difference. In relation to low back pain, the prevalence rates were similar.

Similar results were observed when considering depression, in which the prevalence of neck pain was higher in the physically inactive women when compared to the physically active women. However, the prevalence of low back pain was similar in both groups of patients.

Table 2 presents the associations between anxiety and depression with neck and low back pain according to physical activity levels. In BC survivors classified as inactive, anxiety was significantly associated with neck pain (*p* = 0.030). In women classified as physically active, no association was observed between anxiety and neck pain. Considering the relationship between depression and neck pain, associations were observed in BC survivors classified as inactive, but not in those classified as active. Regarding low back pain, anxiety and depression were not associated, regardless of the level of physical activity of the women BC survivors.

## 4. Discussion

The current study aimed to examine the relationship between anxiety, depression, and neck and back pain in breast cancer survivors, considering the potential influence of physical activity levels. Our findings suggest that higher levels of anxiety and depression are associated with increased neck pain in this population, and that women BC survivors report a high prevalence of neck and low back pain. Additionally, physical activity levels appeared to moderate these relationships, although the effect sizes observed indicate the need for further investigation. Moreover, while the association between anxiety, depression, and neck pain was evident, no statistically significant difference was found for low back pain across physical activity levels, suggesting that different mechanisms may be involved in pain perception at distinct anatomical sites.

The relationship between BC, anxiety, and depression is complex. In this sense, symptoms of anxiety and depression may be associated with episodes of musculoskeletal pain. In a systematic review, Liu et al. [44] observed that symptoms of anxiety and depression were more common in patients with severe neck pain. Similarly, our findings align with this evidence, reinforcing the notion that psychological distress may exacerbate musculoskeletal symptoms in breast cancer survivors. Another study identified significant associations between fatigue, passive shoulder flexion, pressure pain sensitivity, depression, and the intensity of neck and shoulder pain in breast cancer survivors. Depression, neck pain, and passive shoulder flexion were significant predictors of fatigue in this population. These findings suggest that self-reported psychological factors, along with neck pain and shoulder mobility, play a relevant role in cancer-related fatigue [43]. While our study did not directly assess fatigue, the observed associations between anxiety, depression, and neck pain suggest that similar mechanisms could be involved. One factor that may be responsible for this association is that symptoms of anxiety and depression are generally associated with a higher inflammatory profile [45,46], and studies have shown that inflammation is one of the factors linked to a greater chance of experiencing pain [47]. In breast cancer survivors, chronic inflammation resulting from cancer treatments, along with the psychological burden of the disease, could contribute to heightened pain sensitivity [48].

Kim et al. [49] conducted a systematic review of possible factors for the development of neck pain and observed a greatly increased risk of developing neck pain in individuals with high levels of stress and catastrophizing. It is widely known that being diagnosed with BC leads to a high level of stress and negative thoughts like catastrophizing [50,51], which could also explain this association between neck pain and anxiety and depression.

Physical activity has been used as a tool to reduce symptoms of anxiety and depression, as well as episodes of pain. In the current study, the association of anxiety and depression symptoms with cervical pain was observed in the group of women BC survivors classified as physically inactive. Nazari et al. [52] observed that exercises for the cervical flexor muscle reduced pain and episodes of anxiety and depression in patients with chronic neck pain. Christofaro et al. [53], in a study conducted with more than 1800 Brazilian adults, observed that symptoms of depression were associated with musculoskeletal pain, mainly in physically inactive participants. One of the possible mechanisms to explain these findings is the analgesic role that physical activity could play. The different hormones released during physical activity include β-endorphins [54] and serotonin [55], which could contribute to reducing symptoms of anxiety, depression, and, subsequently, episodes of pain [56].

Another relevant aspect is the lack of a significant association between low back pain and physical activity level in our study. This finding contrasts with previous research that showed an inverse relationship between physical activity and musculoskeletal pain [22,57]. Unlike neck pain, which is often associated with stress-related muscle tension and postural imbalances, low back pain may result from structural abnormalities, such as intervertebral disc degeneration, spinal instability, or paraspinal muscle dysfunction [58]. Additionally, some studies suggest that chronic low back pain is more strongly influenced by central sensitization mechanisms rather than peripheral inflammation alone [59]. This could explain why the current study did not find a clear association between physical activity levels and low back pain in breast cancer survivors. Future studies should investigate whether specific types of exercise interventions, such as core stabilization exercises or neuromuscular training, may have a greater impact on low back pain management in this population.

When considering the population of female BC survivors, some important aspects must be considered. A study conducted by Lavallée et al. [60] showed that the debilitating side effects of treatment and the fear of causing irreversible harm serve as barriers to physical activity. Engaging in physical activity within a supportive environment may provide a distraction, allowing individuals to feel “normal” and begin to redefine themselves, shifting their focus toward health and well-being rather than cancer. Providing timely, accurate, and personalized information about the benefits of incorporating physical activity into the treatment regimen, along with guidance from experienced physical activity instructors, may facilitate engagement.

The findings of the current study demonstrated an association that should be considered when promoting interventions for the treatment of these conditions, since the level of physical activity was found to be directly related to symptoms of anxiety, depression, and neck pain [61]. It is also worth highlighting the importance of a multi-disciplinary team, including therapists and specialist nurses working with breast cancer patients, in order to optimize fatigue management. Implementing multidimensional interventions incorporating pain management strategies, mobility exercises, and cognitive behavioral therapy may enhance the patient’s ability to cope with a cancer-related condition [43]. Additionally, individualized exercise prescriptions that consider patient preferences, limitations, and motivational factors could improve adherence and long-term benefits.

Among the main limitations of this study, the observational design prevents cause and effect analyses from being performed. Another point is that the subjective measurement of physical activity is less accurate for identifying the different intensities of physical activity when compared to direct measurement methods, such as accelerometry. Future studies employing objective measures, such as wearable activity trackers or laboratory-based assessments, could provide a more precise evaluation of physical activity levels in breast cancer survivors. Despite this, the current study is pioneering in investigating the relationship between the level of physical activity, anxiety and depression, and musculoskeletal dysfunctions (neck and low back pain), because previous studies have only analyzed these variables in isolation [22,57].

As practical applications of the present study, we emphasize the importance of encouraging the practice of physical activity in female cancer survivors, mainly with the aim of mitigating the relationship between musculoskeletal pain and symptoms of anxiety and depression. Future studies should explore structured physical activity programs tailored for breast cancer survivors, evaluating their effectiveness in controlling symptoms of anxiety and depression and musculoskeletal conditions.

## 5. Conclusions

Our findings revealed a significant association between anxiety, depression, and neck pain specifically in physically inactive women. Conversely, this association was not observed in physically active women, nor was a significant link found between anxiety, depression, and low back pain, irrespective of physical activity level. These results suggest that physical activity may modulate the relationship between psychological distress and neck pain in this population. High prevalence of neck and low back pain was noted, reinforcing their clinical relevance, with the differential findings for neck versus low back pain suggesting distinct underlying mechanisms. The primary contribution of this work is identifying this potential moderating role of physical activity. Practically, these findings underscore the importance of encouraging physical activity among breast cancer survivors to mitigate the link between psychological symptoms and neck pain, highlighting the value of integrated, multidisciplinary care. Study limitations, such as the observational design preventing causal inference and subjective physical activity measurement, should be acknowledged. Future research should employ objective methods and longitudinal designs to clarify causality, exploring targeted interventions, including specific exercise programs.

## Figures and Tables

**Figure 1 biomedicines-13-01089-f001:**
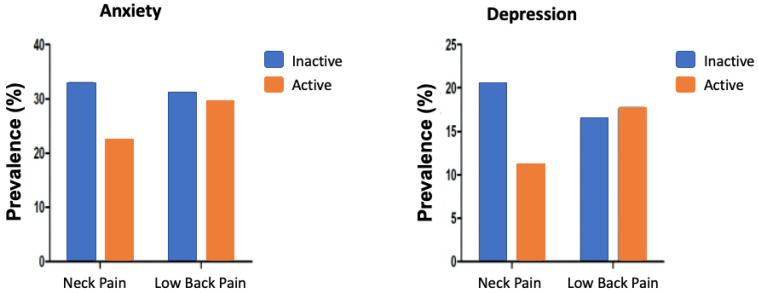
Prevalence of neck and low back pain stratified by anxiety and depression status in physically inactive and active breast cancer survivors.

**Table 1 biomedicines-13-01089-t001:** Characteristics of the sample according to levels of physical activity (*n* = 128).

Variables	Inactive (*n* = 95)	Active (*n* = 33)	*p*
Age (years), Mean (SD)	59.06 (±10.03)	55.88 (±8.58)	0.110
Weight (kg), Mean (SD)	55.10 (±5.13)	54.60 (±6.05)	0.718
Height (cm), Mean (SD)	156.63 (±6.01)	156.72 (±6.39)	0.767
Frequency (%)			
Marital status			
Married, %	70.5	56.3	0.072
Single, %	8.4	9.4	
Divorced, %	13.7	15.6	
Widow, %	7.4	18.8	
Surgery type			
Quadrantectomy, %	52.1	56.3	0.843
Mastectomy, %	47.9	43.8	
Anxiety, %	25.3	21.9	0.882
Depression, %	13.7	18.8	0.683
Neck pain, %	34.7	37.5	0.945
Low back pain, %	57.9	53.1	0.791

**Table 2 biomedicines-13-01089-t002:** Association between anxiety and depression with neck pain and low back pain according to levels of physical activity in BC survivors (*n* = 128).

	OR	95%CI	*p*
	**Neck Pain**		
Anxiety			
Inactive			
Crude model	3.01	1.16–7.38	0.024
Adjusted model	3.10	1.11–8.67	0.030
Active			
Crude model	1.33	0.24–7.33	0.741
Adjusted model	1.15	0.10–13.07	0.909
Depression			
Inactive			
Crude model	3.01	1.16–7.83	0.024
Adjusted model	5.10	1.35–19.27	0.016
Active			
Crude model	1.33	0.24–7.33	0.741
Adjusted model	0.86	0.09–8.24	0.900
	**Low Back pain**		
Anxiety			
Inactive			
Crude model	2.10	0.76–5.71	0.142
Adjusted model	2.37	0.80–7.01	0.117
Active			
Crude model	2.70	0.40–16.88	0.842
Adjusted model	1.87	0.12–29.52	0.655
Depression			
Inactive			
Crude model	1.76	0.50–6.18	0.377
Adjusted model	1.62	0.43–6.04	0.467
Active			
Crude model	0.86	0.14–5.06	0.865
Adjusted model	1.58	0.13–18.72	0.713

Adjusted for age, socioeconomic status, marital status, and type of surgery.

## Data Availability

The original contributions presented in this study are included in the article. Further inquiries can be directed to the corresponding author.

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
