# Peer review of "Physical Inactivity Amplifies the Link Between Anxiety, Depression, and Neck Pain in Breast Cancer Survivors"

_biomedicines, 2025, doi:10.3390/biomedicines13051089_

Round 1
Reviewer 1 Report
Comments and Suggestions for Authors
Dear Authors, I read with interest you manuscript entitled “Association of anxiety and depression with neck pain and low back pain related to physical activity level in women breast cancer survivors”. I believe the topic is very important, in order to understand how to improve patients’ quality of life.
I have some major concerns regarding the English form, which needs a thorough revision, and methodological issues, and some minor comment. Please, find below a detailed list of my suggestions.
-page 2 line 48: Could Authors could give the uptodate incidence and mortality rate?
-page 2, line 65: I suggest the Authors improve the clinical definition of depression, e.g. from the DSM 5.
- page 2 line 72: “due to” instead of “in relation”, which sounds redundant.
-page 2 line 73. What do the Authors mean with “Dignity” here?
-page 2 line 93: I can’t get the meaning of this sentence. Could the Authors rephrase it?
-page 3: it is important to underline that anxiety per se is an emotion, not a pathology; anxiety disorders are pathological. This should be clarified.
- I would suggest a revision of the introduction, in order to focus on the topic of the paper, rather than give general information of different pathologies and situations that distract from the core argument, which are BC survivors. Also, I would review the title in order to make the aim and the results of the study clearer.
-Regarding the methodology, there is no sample size calculation. Is it available?
- Paragraph 2.6 “depressive symptoms”: why the HADS has been used if only depressive scores were used? It would have been more proper to use a dedicated questionnaire, like the Beck Depression Inventory. Also, since anxiety levels are declared to be an endpoint of the study, why they have not been used? No other methods to score anxiety have been declared by the Authors.
- Page 5 line 181: Which cut-off has been used to categorize patients as active or inactive? Based on what rationale?
- Page 5: The Authors state in the methods that only questions regarding depression of the HADS have been used, but in the results, consistently with the title, also anxiety scores are reported. This issue needs clarification.
- Even if the rates of neck and back pain are higher for inactive patients, they are far from being statistically significant, therefore any assumption should be made cautiously.
- Pag 7 lines 215-218: This better fits in the introduction.
- Page 8 line 246: Anxiety is not a pathology, it is an emotion. It would be more proper to use the term “anxiety disorder”.
- Page 8 line 272: It is not the fact that the design is cross-sectional, but rather the fact that it is observational which prevents the conclusions to be causal.
- Page 8: study limitations: no considerations are made regarding the sample size and the power of the study. This should be mentioned in order to gain a critical interpretation of the results.
- Conclusions: the Authors suggest that future studies should focus on protocols to promote physical activity in order to improve musculoskeletal pain and anxiety and depression in BC survivors. Even though I could agree on this on a personal and clinical basis, the study does not allow these causal conclusions; based on these results, future studies should better explore the relationship between musculoskeletal pain and psychological profile in BC survivors, to clarify “what came first, the chicken or the egg”.
Comments on the Quality of English Language
Dear Authors, I read with interest you manuscript entitled “association of anxiety and depression with neck pain and low back pain related to physical activity level in women breast cancer survivors”. I believe the topic is very important, in order to understand how to improve patients’ quality of life.
I have some major concerns regarding the English form, which needs a thorough revision, and methodological issues, and some minor comment. Please, find below a detailed list of my suggestions.
-page 2 line 48: Could Authors could give the uptodate incidence and mortality rate?
-page 2, line 65: I suggest the Authors improve the clinical definition of depression, e.g. from the DSM 5.
- page 2 line 72: “due to” instead of “in relation”, which sounds redundant.
-page 2 line 73. What do the Authors mean with “Dignity” here?
-page 2 line 93: I can’t get the meaning of this sentence. Could the Authors rephrase it?
-page 3: it is important to underline that anxiety per se is an emotion, not a pathology; anxiety disorders are pathological. This should be clarified.
- I would suggest a revision of the introduction, in order to focus on the topic of the paper, rather than give general information of different pathologies and situations that distract from the core argument, which are BC survivors. Also, I would review the title in order to make the aim and the results of the study clearer.
-Regarding the methodology, there is no sample size calculation. Is it available?
- Paragraph 2.6 “depressive symptoms”: why the HADS has been used if only depressive scores were used? It would have been more proper to use a dedicated questionnaire, like the Beck Depression Inventory. Also, since anxiety levels are declared to be an endpoint of the study, why they have not been used? No other methods to score anxiety have been declared by the Authors.
- Page 5 line 181: Which cut-off has been used to categorize patients as active or inactive? Based on what rationale?
- Page 5: The Authors state in the methods that only questions regarding depression of the HADS have been used, but in the results, consistently with the title, also anxiety scores are reported. This issue needs clarification.
- Even if the rates of neck and back pain are higher for inactive patients, they are far from being statistically significant, therefore any assumption should be made cautiously.
- Pag 7 lines 215-218: This better fits in the introduction.
- Page 8 line 246: Anxiety is not a pathology, it is an emotion. It would be more proper to use the term “anxiety disorder”.
- Page 8 line 272: It is not the fact that the design is cross-sectional, but rather the fact that it is observational which prevents the conclusions to be causal.
- Page 8: study limitations: no considerations are made regarding the sample size and the power of the study. This should be mentioned in order to gain a critical interpretation of the results.
- Conclusions: the Authors suggest that future studies should focus on protocols to promote physical activity in order to improve musculoskeletal pain and anxiety and depression in BC survivors. Even though I could agree on this on a personal and clinical basis, the study does not allow these causal conclusions; based on these results, future studies should better explore the relationship between musculoskeletal pain and psychological profile in BC survivors, to clarify “what came first, the chicken or the egg”.
Author Response
Revisor 1 Comentários
Comentários 1) página 2 linha 48: Os autores poderiam fornecer a incidência e a taxa de mortalidade atualizadas?
Resposta 1: Obrigado por apontar isso. Ajustamos esta parte do texto. Por favor, consulte a página 2 ou veja abaixo:
“O câncer de mama (CM) é a neoplasia maligna mais comum entre as mulheres em todo o mundo, com mais de 2 milhões de novos casos e mais de 600.000 mortes por esse câncer, em 2024, ocorridos nos Estados Unidos. (1, 2) ”
Comentários 2) página 2, linha 65: Sugiro que os autores melhorem a definição clínica de depressão, por exemplo, do DSM 5.
Response 2: Thanks for the comment. We adjusted this part of the text. We have included the DSM 5 definition in the revised manuscript. Please refer to the page 2 or see below:
“Depression is described as a mood disorder that causes a persistent feeling of sadness and loss of interest, and which controls patients' attitudes by altering their perception of themselves, so that they start to view situations in their life as major problems. This disorder presents characteristics that may indicate a serious pathology or simply another manifestation of the patient when faced with a real-life situation. (18-21)”
Comments 3) page 2 line 72: “due to” instead of “in relation”, which sounds redundant.
Response 3: Thanks for the comment. We adjusted this part of the text. Please refer to the page 2 or see below:
“The diagnosis of BC and the possible treatments used can generate a great effect on quality of life, which may lead to the development of systemic side effects, perceived shoulder pain, and anxiety and/or depression, due to the stigma related to the pathology and the impacts of the therapy on body appearance and female identity. (10-14)”
Comments 4) page 2 line 73. What do the Authors mean with “Dignity” here?
Response 4: Thanks for the comment. We adjusted this part of the text. Please refer to the page 2 or see below:
“The condition of BC and the development of associated psychological symptoms, tend to lead to decreases in the levels of physical activity in affected individuals, which can contribute to the emergence of musculoskeletal dysfunctions, including neck pain and low back pain. (22)”
Comments 5) page 2 line 93: I can’t get the meaning of this sentence. Could the Authors rephrase it?
Response 5: Thanks for the comment. We have rewritten this part considering the reviewer’s suggestion. Please refer to the pages 2-3 or see below:
“Low back and neck pain are multifactorial diseases and a major problem in modern society, as well as being the leading and fourth leading causes of years lived with disability (YLDs) worldwide, respectively. Moreover, the prevalence of NP is exceeded only by depressive disorders and other musculoskeletal conditions. (30)”
Comments 6) page 3: it is important to underline that anxiety per se is an emotion, not a pathology; anxiety disorders are pathological. This should be clarified.
Response 6: Thanks for the comment. We adjusted this part of the text and improve the definition of anxiety. Please refer to the pages 2-3 or see below:
“Anxiety is an emotional state defined by tension, persistent worry, and physiological responses, such as elevated blood pressure. While anxiety and fear are distinct concepts, they are frequently used interchangeably. Anxiety is regarded as a future-focused and prolonged reaction to a vague or nonspecific threat. (15-17)”
“The gap in the knowledge on the associations of low back and neck pain with anxiety and depression in women who have recovered from BC, demonstrate a clear need for a study on the association of both conditions”
Comments 7) I would suggest a revision of the introduction, in order to focus on the topic of the paper, rather than give general information of different pathologies and situations that distract from the core argument, which are BC survivors. Also, I would review the title in order to make the aim and the results of the study clearer.
Response 7: Thanks for the comment. We adjusted the title and introduction. Please refer to the page 1-2 or see below:
Title suggested by the editor: Physical Inactivity Amplifies the Link Between Anxiety, Depression, and Neck Pain in Breast Cancer Survivors
Comments 8) Regarding the methodology, there is no sample size calculation. Is it available?
Response 8: Thanks for the comment. We adjusted this part of the text. Please refer to the page 3 or see below:
“The sample size estimation was based on a 25% prevalence of breast cancer, considering that this neoplasm accounts for approximately 25% of all cancer cases affecting women, according to data from the National Cancer Institute. (33) A tolerable error of 5% was adopted, resulting in a minimum required sample size of 101 participants. To account for potential sample losses, a 20% increase was applied, establishing a final minimum requirement of 122 participants for this study.”
Comments 9) Paragraph 2.6 “depressive symptoms”: why the HADS has been used if only depressive scores were used? It would have been more proper to use a dedicated questionnaire, like the Beck Depression Inventory. Also, since anxiety levels are declared to be an endpoint of the study, why they have not been used? No other methods to score anxiety have been declared by the Authors.
Response 9: Thanks for the comment. We adjusted this part of the text. Please refer to the page 4 or see below:
“The Hospital Anxiety and Depression Scale (HADS) was used to assess the presence of symptoms of anxiety and depression.(39) The main advantage of the HADS is its validity for use in clinical populations, as it was specifically designed to exclude items that could be generated due to physical conditions rather than psychological states.(40) The HADS questionnaire contains 14 multiple-choice questions, which separately assess symptoms of anxiety (seven questions) and depression (seven questions). Each item receives a score from 0 to 3, with a higher score representing a higher occurrence of anxiety and depression symptoms. The range for each outcome (i.e., anxiety and depression) is from 0 to 21 (41, 42). A score greater than or equal to nine indicates a potential case of anxiety and/or depression. (43) In the present study, the individual item scores were analyzed rather than solely considering the cut-off point for each symptom.”
Comments 10) Page 5 line 181: Which cut-off has been used to categorize patients as active or inactive? Based on what rationale?
Response 10: Thanks for the comment. We adjusted this part of the text. Please refer to the page 4 or see below:
“As the instrument developed by Baecke et al.(34) does not provide specific cut-off points to classify physical activity levels, quartiles were used in the analysis. Women in the highest quartile (quartile 4) were classified as more physically active, while those in the lower quartiles were classified as less active.”
Comments 11) Page 5: The Authors state in the methods that only questions regarding depression of the HADS have been used, but in the results, consistently with the title, also anxiety scores are reported. This issue needs clarification.
Response 11: Thanks for the comment. We adjusted this part of the text. Please refer to the page 4 or see below:
“The Hospital Anxiety and Depression Scale (HADS) was used to assess the presence of symptoms of anxiety and depression.”
Comments 12) Even if the rates of neck and back pain are higher for inactive patients, they are far from being statistically significant, therefore any assumption should be made cautiously.
Response 12: Thanks for the comment. We adjusted this part of the text. Please refer to the page 5 or see below:
“In Figure 1 it can be seen that the prevalence of neck pain was higher in physically inactive patients than in physically active patients, however, there was no statistically significant difference.”
Comments 13) Pag 7 lines 215-218: This better fits in the introduction.
Response 13: Thanks for the comment. We adjusted this part of the text. Please refer to the page 7 or see below:
“The current study aimed to examine the relationship between anxiety, depression, and pain intensity in breast cancer survivors, considering the potential influence of physical activity levels. Our findings suggest that higher levels of anxiety and depression are associated with increased neck pain in this population, and that women BC survivors report a high prevalence of neck and low back pain. Additionally, physical activity levels appeared to moderate these relationships, although the effect sizes observed indicate the need for further investigation. Moreover, while the association between anxiety, depression, and neck pain was evident, no statistically significant difference was found for low back pain across physical activity levels, suggesting that different mechanisms may be involved in pain perception at distinct anatomical sites.”
The relationship between BC, anxiety, and depression is complex. In this sense, symptoms of anxiety and depression may be associated with episodes of musculoskeletal pain.”
Comments 14) Page 8 line 246: Anxiety is not a pathology, it is an emotion. It would be more proper to use the term “anxiety disorder”.
Response 14: Thanks for the comment. We adjusted this part of the text. Please refer to the page 7-8.
Comments 15) Page 8 line 272: It is not the fact that the design is cross-sectional, but rather the fact that it is observational which prevents the conclusions to be causal.
Response 15: Thanks for the comment. We adjusted this part of the text. Please refer to the page 8 or see below:
“Among the main limitations of this study, the observational design prevents cause and effect analyses from being performed.”
Comments 16) Page 8: study limitations: no considerations are made regarding the sample size and the power of the study. This should be mentioned in order to gain a critical interpretation of the results.
Response 16: Thanks for the comment. We adjusted this part of the text. Please refer to the page 3 or see below:
“The sample size estimation was based on a 25% prevalence of breast cancer, considering that this neoplasm accounts for approximately 25% of all cancer cases affecting women, according to data from the National Cancer Institute. (33) A tolerable error of 5% was adopted, resulting in a minimum required sample size of 101 participants. To account for potential sample losses, a 20% increase was applied, establishing a final minimum requirement of 122 participants for this study.”
Comentários 17) Conclusões: os autores sugerem que estudos futuros devem se concentrar em protocolos para promover atividade física a fim de melhorar a dor musculoesquelética, ansiedade e depressão em sobreviventes de CM. Embora eu possa concordar com isso em uma base pessoal e clínica, o estudo não permite essas conclusões causais; com base nesses resultados, estudos futuros devem explorar melhor a relação entre dor musculoesquelética e perfil psicológico em sobreviventes de CM, para esclarecer "o que veio primeiro, a galinha ou o ovo".
Resposta 17: Obrigado pelo comentário. Ajustamos esta parte do texto. Por favor, consulte a página 8 ou veja abaixo:
“Os resultados do estudo atual foram significativos para a associação entre nível de atividade física, ansiedade e depressão, e dor no pescoço em mulheres sobreviventes do câncer de mama. No entanto, a mesma associação não foi encontrada para dor lombar. Estudos futuros devem explorar mais a fundo a relação entre dor musculoesquelética e o perfil psicológico em sobreviventes do câncer de mama para esclarecer qual fator precede o outro.”

Reviewer 2 Report
Comments and Suggestions for Authors
Thank you for the opportunity to review the manuscript " Association of Anxiety and Depression with Neck Pain and Low Back Pain Related to Physical Activity Level in Women Breast Cancer Survivors". The study addresses an important topic regarding the association of anxiety and depression with neck and low back pain in breast cancer survivors. However, there are several areas that require improvement before the manuscript can be considered for publication.
Major Comments:
- The conclusion in the abstract needs to be rewritten to ensure clarity and coherence. When read independently, it does not fully convey the study’s findings in a meaningful way.
- Please ensure that all references follow the journal's required format.
- Provide a clear justification for using the Hospital Anxiety and Depression Scale (HADS) for mood assessment. Explain why this tool was chosen over alternative scales and how it aligns with the study objectives.
- Insert the "±" symbol in tables to represent standard deviations (SD). This improves readability and ensures consistency with statistical conventions.
- The images currently presented are of low quality. Improve their resolution and ensure that SD bars are included.
- Include an analysis of the statistical power of the sample to confirm whether the study has adequate power to detect meaningful associations.
- Explicitly acknowledge the small sample size as a limitation in the discussion section and address how this may affect the generalizability of findings.
Minor Comments:
The manuscript contains several grammatical errors that need to be addressed. Here are a few examples among many:
- Incorrect article usage
- "Breast Cancer is the most frequent malignant neoplasm in women in worldwide with a growing incidence."
- Correction: "Breast cancer is the most frequent malignant neoplasm in women worldwide, with a growing incidence."
- Issue: "in worldwide" is incorrect. "Worldwide" does not need "in."
- Sentence fragments or awkward phrasing
- "The global breast cancer mortality rate has increased by 0.23% per year, with statistically significant rises observed in age groups under 50 and those aged 70 and older."
- Correction: "The global breast cancer mortality rate has increased by 0.23% per year, with statistically significant rises observed in individuals under 50 and those 70 and older."
- Issue: "age groups under 50 and those aged 70 and older" is awkward and redundant.
- Verb agreement errors
- "It is widely that physical activity is one tool has contributed to improving symptoms of anxiety and depression and reducing musculoskeletal pain is physical activity."
- Correction: "It is widely recognized that physical activity contributes to improving symptoms of anxiety and depression and reducing musculoskeletal pain."
- Issue: Incorrect sentence structure and missing verb agreement.
- Misuse of prepositions
- "The study was conducted with 128 women BC survivors who were invited to participate through a telephone call from the BC Support Association of Presidente Prudente."
- Correction: "The study was conducted with 128 female breast cancer survivors who were invited to participate via a telephone call from the BC Support Association of Presidente Prudente."
- Issue: "with" should be "among" or "on," and "through a telephone call" should be "via a telephone call."
- Redundancies and wordiness
- "As the main findings of this study, we observed that the prevalence of neck and low back pain was high in women who survived BC."
- Correction: "Our main finding was a high prevalence of neck and low back pain in breast cancer survivors."
- Issue: The phrase "As the main findings of this study, we observed that" is unnecessarily wordy.
- Unclear pronoun references
- "In this way, can be identified the best therapeutic conduct to promote biopsychosocial well-being."
- Correction: "In this way, the best therapeutic approach to promote biopsychosocial well-being can be identified."
- Issue: "Can be identified" is misplaced.
- Improper verb forms
- "To assess the presence of low back pain and neck pain was used the Nordic questionnaire."
- Correction: "The Nordic questionnaire was used to assess the presence of low back and neck pain."
- Issue: Incorrect verb order.
The manuscript contains several grammatical errors that need to be addressed. Here are a few examples among many:
- Incorrect article usage
- "Breast Cancer is the most frequent malignant neoplasm in women in worldwide with a growing incidence."
- Correction: "Breast cancer is the most frequent malignant neoplasm in women worldwide, with a growing incidence."
- Issue: "in worldwide" is incorrect. "Worldwide" does not need "in."
- Sentence fragments or awkward phrasing
- "The global breast cancer mortality rate has increased by 0.23% per year, with statistically significant rises observed in age groups under 50 and those aged 70 and older."
- Correction: "The global breast cancer mortality rate has increased by 0.23% per year, with statistically significant rises observed in individuals under 50 and those 70 and older."
- Issue: "age groups under 50 and those aged 70 and older" is awkward and redundant.
- Verb agreement errors
- "It is widely that physical activity is one tool has contributed to improving symptoms of anxiety and depression and reducing musculoskeletal pain is physical activity."
- Correction: "It is widely recognized that physical activity contributes to improving symptoms of anxiety and depression and reducing musculoskeletal pain."
- Issue: Incorrect sentence structure and missing verb agreement.
- Misuse of prepositions
- "The study was conducted with 128 women BC survivors who were invited to participate through a telephone call from the BC Support Association of Presidente Prudente."
- Correction: "The study was conducted with 128 female breast cancer survivors who were invited to participate via a telephone call from the BC Support Association of Presidente Prudente."
- Issue: "with" should be "among" or "on," and "through a telephone call" should be "via a telephone call."
- Redundancies and wordiness
- "As the main findings of this study, we observed that the prevalence of neck and low back pain was high in women who survived BC."
- Correction: "Our main finding was a high prevalence of neck and low back pain in breast cancer survivors."
- Issue: The phrase "As the main findings of this study, we observed that" is unnecessarily wordy.
- Unclear pronoun references
- "In this way, can be identified the best therapeutic conduct to promote biopsychosocial well-being."
- Correction: "In this way, the best therapeutic approach to promote biopsychosocial well-being can be identified."
- Issue: "Can be identified" is misplaced.
- Improper verb forms
- "To assess the presence of low back pain and neck pain was used the Nordic questionnaire."
- Correction: "The Nordic questionnaire was used to assess the presence of low back and neck pain."
- Issue: Incorrect verb order.
Author Response
Reviewer Comments 2
Major Comments:
Comments 1) The conclusion in the abstract needs to be rewritten to ensure clarity and coherence. When read independently, it does not fully convey the study’s findings in a meaningful way.
Response 1: Thanks for the comment. We adjusted this part of the text. Please refer to page 1 or see below:
“This study demonstrates significant results for the association between physical activity level, anxiety and depression, and neck pain in women BC survivors. However, the same association was not found for low back pain.”
Comments 2) Please ensure that all references follow the journal's required format.
Response 2: Thanks for the comment. We have reviewed all references. Please refer to page 9.
Comments 3) Provide a clear justification for using the Hospital Anxiety and Depression Scale (HADS) for mood assessment. Explain why this tool was chosen over alternative scales and how it aligns with the study objectives.
Response 3: Thanks for the comment. We adjusted this part of the text. Please refer to page 4 or see below:
“The Hospital Anxiety and Depression Scale (HADS) was used to assess the presence of symptoms of anxiety and depression.(39) The main advantage of the HADS is its validity for use in clinical populations, as it was specifically designed to exclude items that could be generated due to physical conditions rather than psychological states.(40)”
Comments 4) Insert the "±" symbol in tables to represent standard deviations (SD). This improves readability and ensures consistency with statistical conventions.
Response 4: Thanks for the comment. We adjusted this part of the text. Please refer to page 5.
Comments 5) The images currently presented are of low quality. Improve their resolution and ensure that SD bars are included.
Response 5: Thanks for the comment. We improved the image, and it is not possible to insert SD bars because is a prevalence graph.
Comments 6) Include an analysis of the statistical power of the sample to confirm whether the study has adequate power to detect meaningful associations.
Response 6: Thanks for the comment. We adjusted this part of the text. Please refer to page 3 or see below:
“The sample size estimation was based on a 25% prevalence of breast cancer, considering that this neoplasm accounts for approximately 25% of all cancer cases affecting women, according to data from the National Cancer Institute. (33) A tolerable error of 5% was adopted, resulting in a minimum required sample size of 101 participants. To account for potential sample losses, a 20% increase was applied, establishing a final minimum requirement of 122 participants for this study.”
Comments 7) Explicitly acknowledge the small sample size as a limitation in the discussion section and address how this may affect the generalizability of findings.
Response 7: Thanks for the comment. We adjusted this part of the text. We included information about sample size. Please refer to page or see below:
“The sample size estimation was based on a 25% prevalence of breast cancer, considering that this neoplasm accounts for approximately 25% of all cancer cases affecting women, according to data from the National Cancer Institute. (33) A tolerable error of 5% was adopted, resulting in a minimum required sample size of 101 participants. To account for potential sample losses, a 20% increase was applied, establishing a final minimum requirement of 122 participants for this study.”
Minor Comments:
The manuscript contains several grammatical errors that need to be addressed. Here are a few examples among many:
Incorrect article usage
Comments 8) "Breast Cancer is the most frequent malignant neoplasm in women in worldwide with a growing incidence."
Correction: "Breast cancer is the most frequent malignant neoplasm in women worldwide, with a growing incidence."
Issue: "in worldwide" is incorrect. "Worldwide" does not need "in."
Response 8: Thanks for the comment. We adjusted this part of the text. Please refer to page 1 or see below:
“Breast cancer (BC) is the most common malignant neoplasm among women worldwide, with over 2 million new cases, and more than 600,000 deaths from this cancer, in 2024 in the United States.” (1, 2)
Sentence fragments or awkward phrasing
Comments 9)"The global breast cancer mortality rate has increased by 0.23% per year, with statistically significant rises observed in age groups under 50 and those aged 70 and older."
Correction: "The global breast cancer mortality rate has increased by 0.23% per year, with statistically significant rises observed in individuals under 50 and those 70 and older."
Issue: "age groups under 50 and those aged 70 and older" is awkward and redundant.
Response 9: Thanks for the comment. We adjusted this part of the text. Please refer to page 2 or see below:
“The global breast cancer mortality rate has increased by 0.23% per year, with statistically significant rises observed in individuals under 50 and over 70 years of age.” (4)
Verb agreement errors
Comments 10) "It is widely that physical activity is one tool has contributed to improving symptoms of anxiety and depression and reducing musculoskeletal pain is physical activity."
Correction: "It is widely recognized that physical activity contributes to improving symptoms of anxiety and depression and reducing musculoskeletal pain."
Issue: Incorrect sentence structure and missing verb agreement.
Response 10: Thanks for the comment. We adjusted this part of the text. Please refer to page 3 or see below:
“It is widely recognized that physical activity contributes to improving symptoms of anxiety and depression and reducing musculoskeletal pain. (31)”
Misuse of prepositions
Comments 11) "The study was conducted with 128 women BC survivors who were invited to participate through a telephone call from the BC Support Association of Presidente Prudente."
Correction: "The study was conducted with 128 female breast cancer survivors who were invited to participate via a telephone call from the BC Support Association of Presidente Prudente."
Issue: "with" should be "among" or "on," and "through a telephone call" should be "via a telephone call."
Response 11: Thanks for the comment. We adjusted this part of the text. Please refer to page 3 or see below:
“The study was conducted with 128 female breast cancer survivors. Participant recruitment was conducted through institutions that support breast cancer and referrals from mastologists in the city. Data collection was carried out through face-to-face interviews conducted by previously trained researchers.”
Redundancies and wordiness
Comments 12) "As the main findings of this study, we observed that the prevalence of neck and low back pain was high in women who survived BC."
Correction: "Our main finding was a high prevalence of neck and low back pain in breast cancer survivors."
Issue: The phrase "As the main findings of this study, we observed that" is unnecessarily wordy.
Response 12: Thanks for the comment. We adjusted this part of the text. Please refer to page 7 or see below:
“The current study aimed to examine the relationship between anxiety, depression, and pain intensity in breast cancer survivors, considering the potential influence of physical activity levels. Our findings suggest that higher levels of anxiety and depression are associated with increased neck pain in this population, and that women BC survivors report a high prevalence of neck and low back pain.”
Unclear pronoun references
Comments 13) "In this way, can be identified the best therapeutic conduct to promote biopsychosocial well-being."
Correction: "In this way, the best therapeutic approach to promote biopsychosocial well-being can be identified."
Issue: "Can be identified" is misplaced.
Response 13: Thanks for the comment. We adjusted this part of the text. Please refer to page 3 or see below:
“In this way, the results can help to identify the best therapeutic approach to promote biopsychosocial well-being.”
Improper verb forms
Comments 14) "To assess the presence of low back pain and neck pain was used the Nordic questionnaire."
Correction: "The Nordic questionnaire was used to assess the presence of low back and neck pain."
Issue: Incorrect verb order.
Response 14: Thanks for the comment. We adjusted this part of the text. Please refer to page 4 or see below:
“NP and LBP were assessed using the Nordic Musculoskeletal Questionnaire, designed to report musculoskeletal symptoms.(44)”
Reviewer 3 Report
Comments and Suggestions for Authors
The topic of the manuscript "Association of Anxiety and Depression with Neck Pain and Low Back Pain Related to Physical Activity Level in Women Breast Cancer Survivors" is interesting, but the manuscript should be revised as the submitted document is very confusing. The main problems are detailed below.
-It is not clear how anxiety was assessed. Specifically, on page 4, lines 153 to 155, the following is stated:
“2.6. Depressive symptoms
The Hospital Anxiety and Depression Scale (HADS) was used to assess the presence of symptoms of depression.” And on lines 159-161 they add: “In this study, only questions related to depressive symptoms were considered, with a score ≥ 9 corresponding to the presence of symptoms of depression”.
Furthermore, it should be made clear throughout the manuscript whether it is the symptoms of depression and anxiety or the disorder, because in lines 103-104, referring to anxiety and depression, they say “to understand if there is an association between the afore mentioned diseases and the decrease in the practice of physical activities”.
- The title and abstract refer to “physical activity level”, but Table 1 and Table 2 refer only to "Active" or "Inactive", so this should be the terminology used.
Abstract
On page 1, lines 33 and 34 it says: “The presence of anxiety, depression, and neck pain was higher in physically inactive women than in active women.” However, according to Table 1, there are no statistically significant differences in anxiety, depression and neck pain between Inactive and Active women.
Introduction
The Introduction section should be revised. The impact of the disease on women (page 2 lines 59 to 76) should be thoroughly revised as this part is rather confusing and incomplete in general. A review of what has been published should be carried out and based on the results found, it should be rewritten as it currently seems very partial and biased. In general, in the Introduction section, each specific topic should not be based on a single work, but on several published works. For example, in lines 65 to 69 it says: “Depression is described as a mood disorder, which controls patients' attitudes by altering their perception of themselves, starting to view their situations as major problems. It has characteristics that may indicate a serious pathology or be just another manifestation of the patient in front of a real-life situation. (14)”. Such a conceptualisation does not seem to be the most appropriate when it comes to breast cancer survivors, many of whom may have depressive symptoms but not a disorder with the characteristics mentioned. Lines 70 to 73 read: “The diagnosis of BC and the possible treatments to which the patient will be exposed generate a great psychological shock, which can lead to the development of anxiety and/or depression, in relation to the stigma related to the pathology and the impacts of the therapy on body appearance and female identity. (10)”. Such a statement should not be based on a single paper but on the work of several authors and should be worded in such a way as to make it clear that it refers to all the variables mentioned (e.g. “psychological shock”, “female identity”) as well as to the proposed causal relationships. Lines 73 to 75 read: “With the dignity of BC and the development of associated mental disorders, they tend to decrease their levels of physical activity, and this can contribute to the emergence of musculoskeletal dysfunctions, including neck pain and low back pain. (15).” The statement "BC and the development of associated mental disorders" is a claim that should be supported by several research, because the fact that women with breast cancer develop some psychological symptoms does not mean that breast cancer is associated with “mental disorders”.
Methods
-The procedure for accessing the sample should be clearly and fully explained. There should also be clear and explicit information on how “women BC survivors” are defined.
- The assessment of physical activity should be better explained, clearly stating the criteria used to define the categories of "Active" and "Inactive".
- The assessment of musculoskeletal pain should be thoroughly reviewed as it is very confusing.
- The way in which anxiety was assessed should be included.
Results
Figure 1 should be revised and it should be explained what exactly "Anxiety" and "Depression" mean.
Discussion
On page 8, lines 256 to 259 it says: “Our study shows a relationship that deserves attention to promote interventions for the treatment of these conditions, since the level of physical activity is directly related to symptoms of anxiety, depression, and neck and back pain, considering patients' preferences. (50)” This text should be revised because, according to Table 1, there are no statistically significant differences in anxiety, depression, and neck and back pain between active and inactive women. The conclusion should also be revised as it is inconsistent with these results.
Comments on the Quality of English LanguageThe English language should be revised. For example, on page 2, lines 51 and 52, it says “and most of the number of cases is 100 times higher in women than in men”.
Author Response
Reviewer comments 3
Comments 1) It is not clear how anxiety was assessed. Specifically, on page 4, lines 153 to 155, the following is stated:
“2.6. Depressive symptoms
Response 1: Thanks for the comment. We adjusted this part of the text. Please refer to the page 4 or see below:
“The Hospital Anxiety and Depression Scale (HADS) was used to assess the presence of symptoms of anxiety and depression.(39) The main advantage of the HADS is its validity for use in clinical populations, as it was specifically designed to exclude items that could be generated due to physical conditions rather than psychological states.(40) The HADS questionnaire contains 14 multiple-choice questions, which separately assess symptoms of anxiety (seven questions) and depression (seven questions). Each item receives a score from 0 to 3, with a higher score representing a higher occurrence of anxiety and depression symptoms. The range for each outcome (i.e., anxiety and depression) is from 0 to 21 (41, 42). A score greater than or equal to nine indicates a potential case of anxiety and/or depression. (43) In the present study, the individual item scores were analyzed rather than solely considering the cut-off point for each symptom.”
Comments 2) The Hospital Anxiety and Depression Scale (HADS) was used to assess the presence of symptoms of depression.” And on lines 159-161 they add: “In this study, only questions related to depressive symptoms were considered, with a score ≥ 9 corresponding to the presence of symptoms of depression”.
Furthermore, it should be made clear throughout the manuscript whether it is the symptoms of depression and anxiety or the disorder, because in lines 103-104, referring to anxiety and depression, they say “to understand if there is an association between the afore mentioned diseases and the decrease in the practice of physical activities”.
Response 2: Thanks for the comment. We adjusted this part of the text. Please refer to the page 3 and 4 or see below:
“The current work differs from previous studies, as it aims to understand if there is an association between the aforementioned symptoms and the decrease in the practice of physical activities.”
“The Hospital Anxiety and Depression Scale (HADS) was used to assess the presence of symptoms of anxiety and depression.(39) The main advantage of the HADS is its validity for use in clinical populations, as it was specifically designed to exclude items that could be generated due to physical conditions rather than psychological states.(40) The HADS questionnaire contains 14 multiple-choice questions, which separately assess symptoms of anxiety (seven questions) and depression (seven questions). Each item receives a score from 0 to 3, with a higher score representing a higher occurrence of anxiety and depression symptoms. The range for each outcome (i.e., anxiety and depression) is from 0 to 21 (41, 42). A score greater than or equal to nine indicates a potential case of anxiety and/or depression. (43) In the present study, the individual item scores were analyzed rather than solely considering the cut-off point for each symptom.”
Comments 3) The title and abstract refer to “physical activity level”, but Table 1 and Table 2 refer only to "Active" or "Inactive", so this should be the terminology used.
Response 3: Thanks for the comment. We adjusted the title and abstract. Please refer to the page 1 or see below:
Title suggested by the editor: Physical Inactivity Amplifies the Link Between Anxiety, Depression, and Neck Pain in Breast Cancer Survivors
“To analyze the relationship women breast cancer survivors physically active and inactive with incidence of neck pain and low back pain, in association with anxiety and depression.”
Abstract
Comments 4) On page 1, lines 33 and 34 it says: “The presence of anxiety, depression, and neck pain was higher in physically inactive women than in active women.” However, according to Table 1, there are no statistically significant differences in anxiety, depression and neck pain between Inactive and Active women.
Response 4: Thank you for pointing this out. We adjusted this part of the text. Please refer to the page 1 or see below:
“The presence of anxiety, depression, and neck pain was higher in physically inactive women than in active women, however, there was no statistically significant difference.”
Introduction
Comments 5) The Introduction section should be revised. The impact of the disease on women (page 2 lines 59 to 76) should be thoroughly revised as this part is rather confusing and incomplete in general. A review of what has been published should be carried out and based on the results found, it should be rewritten as it currently seems very partial and biased. In general, in the Introduction section, each specific topic should not be based on a single work, but on several published works. For example, in lines 65 to 69 it says: “Depression is described as a mood disorder, which controls patients' attitudes by altering their perception of themselves, starting to view their situations as major problems. It has characteristics that may indicate a serious pathology or be just another manifestation of the patient in front of a real-life situation. (14)”. Such a conceptualisation does not seem to be the most appropriate when it comes to breast cancer survivors, many of whom may have depressive symptoms but not a disorder with the characteristics mentioned. Lines 70 to 73 read: “The diagnosis of BC and the possible treatments to which the patient will be exposed generate a great psychological shock, which can lead to the development of anxiety and/or depression, in relation to the stigma related to the pathology and the impacts of the therapy on body appearance and female identity. (10)”. Such a statement should not be based on a single paper but on the work of several authors and should be worded in such a way as to make it clear that it refers to all the variables mentioned (e.g. “psychological shock”, “female identity”) as well as to the proposed causal relationships. Lines 73 to 75 read: “With the dignity of BC and the development of associated mental disorders, they tend to decrease their levels of physical activity, and this can contribute to the emergence of musculoskeletal dysfunctions, including neck pain and low back pain. (15).” The statement "BC and the development of associated mental disorders" is a claim that should be supported by several research, because the fact that women with breast cancer develop some psychological symptoms does not mean that breast cancer is associated with “mental disorders”.
Response 5: We have rewritten the Introduction considering the reviewer’s suggestion. Please refer to the page 2.
Methods
Comments 6) The procedure for accessing the sample should be clearly and fully explained. There should also be clear and explicit information on how “women BC survivors” are defined.
Response 6: Thanks for the comment. We adjusted this part of the text. Please refer to the page 3 or see below:
“The study was conducted with 128 female breast cancer survivors. Participant recruitment was conducted through institutions that support breast cancer and referrals from mastologists in the city. Data collection was carried out through face-to-face interviews conducted by previously trained researchers.
All participants provided written informed consent, voluntarily agreeing to participate in the study, after being fully informed about the research procedures and objectives.
The selected inclusion criteria were women who had survived breast cancer and being over 18 years of age. The exclusion factors were having a pacemaker or metal plate, due to bioelectrical impedance, or any type of mental disorder that would prevent the woman from answering the questionnaires.”
Comments 7) The assessment of physical activity should be better explained, clearly stating the criteria used to define the categories of "Active" and "Inactive".
Response 7: Thanks for the comment. We adjusted this part of the text. Please refer to the page 4 or see below:
“As the instrument developed by Baecke et al.(34) does not provide specific cut-off points to classify physical activity levels, quartiles were used in the analysis. Women in the highest quartile (quartile 4) were classified as more physically active, while those in the lower quartiles were classified as less active.”
Comments 8) The assessment of musculoskeletal pain should be thoroughly reviewed as it is very confusing.
Response 8: Thanks for the comment. We adjusted this part of the text. Please refer to the page 4 or see below:
“NP and LBP were assessed using the Nordic Musculoskeletal Questionnaire, designed to report musculoskeletal symptoms.(44) The questionnaire is divided into two parts: the first part is a general survey where the participant indicates the body part(s) in which they experience pain. The second part is a specific questionnaire where the participant answers questions about the duration of symptoms throughout their life, in the previous 12 months, and in the previous 7 days. In addition to symptom duration, the questionnaire also assesses the impact of pain on work and leisure activities, the presence of medical leave, and symptom duration.”
Comments 9) The way in which anxiety was assessed should be included.
Response 9: Thanks for the comment. We adjusted this part of the text. Please refer to the page 4 or see below:
“The Hospital Anxiety and Depression Scale (HADS) was used to assess the presence of symptoms of anxiety and depression.(39) The main advantage of the HADS is its validity for use in clinical populations, as it was specifically designed to exclude items that could be generated due to physical conditions rather than psychological states.(40) The HADS questionnaire contains 14 multiple-choice questions, which separately assess symptoms of anxiety (seven questions) and depression (seven questions). Each item receives a score from 0 to 3, with a higher score representing a higher occurrence of anxiety and depression symptoms. The range for each outcome (i.e., anxiety and depression) is from 0 to 21 (41, 42). A score greater than or equal to nine indicates a potential case of anxiety and/or depression. (43) In the present study, the individual item scores were analyzed rather than solely considering the cut-off point for each symptom.”
Results
Comments 10) Figure 1 should be revised and it should be explained what exactly "Anxiety" and "Depression" mean.
Response 10: Thanks for the comment. We revised and improved the explaine about anxiety and depression. Please refer to the page 4 or see below:
“The Hospital Anxiety and Depression Scale (HADS) was used to assess the presence of symptoms of anxiety and depression.(39) The main advantage of the HADS is its validity for use in clinical populations, as it was specifically designed to exclude items that could be endorsed due to physical conditions rather than psychological states.(40) The HADS questionnaire contains 14 multiple-choice questions, which separately assess symptoms of anxiety (seven questions) and depression (seven questions). Each item scored from 0 to 3, with a higher score representing a higher occurrence of anxiety and depression symptoms. The range for each outcome (i.e., anxiety and depression) is from 0 to 21 (41, 42). A score greater than or equal to nine indicates a potential case of anxiety and/or depression. (43) In the present study, the individual item scores were analyzed rather than solely considering the cut-off point for each symptom.”
Discussion
Comments 11) On page 8, lines 256 to 259 it says: “Our study shows a relationship that deserves attention to promote interventions for the treatment of these conditions, since the level of physical activity is directly related to symptoms of anxiety, depression, and neck and back pain, considering patients' preferences. (50)” This text should be revised because, according to Table 1, there are no statistically significant differences in anxiety, depression, and neck and back pain between active and inactive women. The conclusion should also be revised as it is inconsistent with these results.
Response 11: Thanks for the comment. We adjusted this part of the text. Please refer to the page 10 or see below:
“The results of the current study were significant for the association between physical activity level, anxiety and depression, and neck pain in women BC survivors. However, the same association was not found for low back pain. Future studies should further explore the relationship between musculoskeletal pain and the psychological profile in breast cancer survivors in order to clarify which factor precedes the other. “
Comments 12) The English language should be revised. For example, on page 2, lines 51 and 52, it says “and most of the number of cases is 100 times higher in women than in men”.
Response 12: Thanks for the comment. We reviewed the entire text with an English native speaker.
Round 2
Reviewer 1 Report
Comments and Suggestions for Authors
Dear Authors,
thank you for answering my comments.
A couple of your answers are written in your native language, therefore I caught something, but not everything.
I am still concerned for the sample size calculation that you provided, because the endpoints are not clear (you state a 25% prevalence of breast cancer, which, however, defines your population but not your outcome). Could you give more details of the power analysis?
Thank you.
Comments on the Quality of English LanguageThe english form is still not optimal, some senteces are tricky and hard to follow. I would suggest a professional revision of the quality fo english.
Author Response
Reviewer 1
Comments 1: I am still concerned for the sample size calculation that you provided, because the endpoints are not clear (you state a 25% prevalence of breast cancer, which, however, defines your population but not your outcome). Could you give more details of the power analysis?
Response 1: Dear reviewer, thank you for your comment. Based on your comment, we adjusted the sample size calculation by investigating the relationship between anxiety and depression symptoms and back pain, which are the main results of our study. To this end, we used the study by Hung et al., in which a correlation between anxiety symptoms and low back pain of r=0.30 was observed. Considering an alpha error of 5% and a sample power of 80%, plus 20% related to sample loss, the minimum number of participants for the present study is 101 participants. Please refer to the page 3 or see below:
“To calculate the sample, a correlation of r=0.30 was considered between anxiety symptoms and low back pain based on the study by Hung et al., with an alpha error of 5% and a sample power of 80%. Anticipating possible losses, an additional 20% was added to the sample size, requiring a minimum of 101 participants. (33)”
Comments on the Quality of English Language
Comments 2: The English form is still not optimal, some senteces are tricky and hard to follow. I would suggest a professional revision of the quality fo english.
Response 2: Thanks for the comment. We reviewed the entire text with an English native speaker.

Reviewer 2 Report
Comments and Suggestions for Authors
The authors have made all the requested modifications and improvements, and therefore, in my opinion, the article is ready for publication as long as it follows the journal's technical recommendations
Author Response
Reviewer 2
Comments 1: The authors have made all the requested modifications and improvements, and therefore, in my opinion, the article is ready for publication as long as it follows the journal's technical recommendations
Response 1: Thanks for the comment.
Reviewer 3 Report
Comments and Suggestions for Authors
The revised manuscript “Physical Inactivity Amplifies the Link Between Anxiety, Depression, and Neck Pain in Breast Cancer Survivors” is an improvement over the first version, but it is still extremely confusing. Some of the deficiencies found are detailed below.
-In the Abstract, lines 25 to 27 say “With this in mind, the aim of the current study was to analyze the relationship of women breast cancer survivors, physically active and physically inactive, with the incidences of neck pain and low back pain, in association with anxiety and depression.” This sentence should be revised and the aim of the study should be clearly stated.
-In the Abstract, on lines 28 to 30, it says: “The presence of anxiety, depression, and neck pain was higher in physically inactive women than in active women, however, there was no statistically significant difference.”. But according to the data in Table 1, the score for depression is higher for active women (18.8) than for inactive women (13.7), and the same is true for neck pain, which is higher for active women (37.5) than for inactive women (34.7).
- On page 2, line 86, it says "prevalence of NP is". It should be explained what "NP" means since it appears for the first time.
- On page 3, lines 106-107 it says: “All volunteers who agreed to participate in the study signed an informed consent form”. And on the same page, lines 119-120, it says “All participants provided written informed consent, voluntarily agreeing to participate in the study”. The same information should not be repeated.
-On page 4, lines 151 to 153, it says: “Women in the highest quartile (quartile 4) were classified as more physically active, while those in the lower quartiles were classified as less active”. However, in Table 1 and Figure 1 it says "inactive" and "active". It should be revised to use the correct terms. In addition, it should report the number of women classified in each activity category, as well as the range of total activity scores and the specific cut-off point for classification.
-Although the description of the musculoskeletal pain assessment has been improved, it is still unclear what the scores that appear in the manuscript mean. Specifically, on page 5, lines 180 to 183, it states, “participant answers questions about the duration of symptoms throughout their life, in the previous 12 months, and in the previous 7 days. In addition to symptom duration, the questionnaire also assesses the impact of pain on work and leisure activities, the presence of medical leave, and symptom duration”. Which of these possible scores are used in the present manuscript? What exactly do the low back and the neck pain scores in Table 1 mean? Figure 1 refers to "prevalence rates of neck and low back pain". How was prevalence computed?
- In Table 1, it should be clear what the scores refer to, specifying when they are means and when they are percentages or rates. In addition, the number of women in each of the two categories should be added.
-Figure 1 is also confusing. The caption of the figure (on page 6, lines 211,212) reads: “Figure 1. Prevalence rates of neck and low back pain according to anxiety and depression in Inactive and Active BC Survivors”. What exactly does “according to anxiety and depression” mean?
-Throughout the manuscript, it should be completely clear when referring to “rates” and when referring to “symptoms” or “intensity”. For example, Figure 1 states that it refers to prevalence rates of neck and low back pain, but the discussion refers to pain intensity. Specifically, on page 7, lines 225-226, it says: “The current study aimed to examine the relationship between anxiety, depression, and pain intensity”. It is suggested that the entire manuscript, including the Discussion, be thoroughly reviewed.
Author Response
Reviewer 3
Comments 1: In the Abstract, lines 25 to 27 say “With this in mind, the aim of the current study was to analyze the relationship of women breast cancer survivors, physically active and physically inactive, with the incidences of neck pain and low back pain, in association with anxiety and depression.” This sentence should be revised and the aim of the study should be clearly stated.
Response 1: Thanks for the comment. We adjusted this part of the text. Please refer to the page 1 or see below:
“With this in mind, the aim of the present study was to analyze the relationship between physical activity levels (active vs. inactive), neck and low back pain in women breast cancer survivors in association with anxiety and depression”
Comments 2: In the Abstract, on lines 28 to 30, it says: “The presence of anxiety, depression, and neck pain was higher in physically inactive women than in active women, however, there was no statistically significant difference.”. But according to the data in Table 1, the score for depression is higher for active women (18.8) than for inactive women (13.7), and the same is true for neck pain, which is higher for active women (37.5) than for inactive women (34.7).
Response 2: Thanks for the comment. Dear Reviewer, the information in the abstract regarding prevalence data refers to Figure 1, which presents a comparison among women with anxiety and depression.
Comments 3: On page 2, line 86, it says "prevalence of NP is". It should be explained what "NP" means since it appears for the first time.
Response 3: Thanks for the comment. We adjusted this part of the text. Please refer to the page 2 or see below:
“Moreover, the prevalence of neck pain (NP) is exceeded only by depressive disorders and other musculoskeletal conditions. (30)”
Comments 4: On page 3, lines 106-107 it says: “All volunteers who agreed to participate in the study signed an informed consent form”. And on the same page, lines 119-120, it says “All participants provided written informed consent, voluntarily agreeing to participate in the study”. The same information should not be repeated.
Response 4: Thanks for the comment. We adjusted this part of the text. Please refer to the page 3.
Comments 5: On page 4, lines 151 to 153, it says: “Women in the highest quartile (quartile 4) were classified as more physically active, while those in the lower quartiles were classified as less active”. However, in Table 1 and Figure 1 it says "inactive" and "active". It should be revised to use the correct terms. In addition, it should report the number of women classified in each activity category, as well as the range of total activity scores and the specific cut-off point for classification.
Response 5: Dear reviewer, thank you for your comment. We have adjusted the text and, to make it clearer, we have considered the classification as active and inactive. In Table 1, we have included the number of women as active (n=33) and as inactive (n=95). Based on the Baecke questionnaire, we have added in the manuscript, or specific cutoff point, as required. Please refer to the page 4 and 5
Comments 6: Although the description of the musculoskeletal pain assessment has been improved, it is still unclear what the scores that appear in the manuscript mean. Specifically, on page 5, lines 180 to 183, it states, “participant answers questions about the duration of symptoms throughout their life, in the previous 12 months, and in the previous 7 days. In addition to symptom duration, the questionnaire also assesses the impact of pain on work and leisure activities, the presence of medical leave, and symptom duration”. Which of these possible scores are used in the present manuscript? What exactly do the low back and the neck pain scores in Table 1 mean? Figure 1 refers to "prevalence rates of neck and low back pain". How was prevalence computed?
Response 6: Dear reviewer, the Nordic questionnaire has yes or no response options. In the present study, we adopted the report of pain in the last seven days. The prevalence was computed through the participants who answered yes to the presence of neck and/or back pain in the last week. This information was added to the manuscript to make it clearer to readers. Please refer to the page 4 or see below:
“In addition to symptom duration, the questionnaire also assesses the impact of pain on work and leisure activities, the presence of medical leave, and symptom duration. In this instrument, there are two possible answers: yes or no. In the present study, responses reporting cervical and/or lumbar pain in the last seven days were considered.”
Comments 7: In Table 1, it should be clear what the scores refer to, specifying when they are means and when they are percentages or rates. In addition, the number of women in each of the two categories should be added.
Response 7: Thanks for the comment. We adjusted this part of the text. Please refer to the page 5.
Response 8: Figure 1 is also confusing. The caption of the figure (on page 6, lines 211,212) reads: “Figure 1. Prevalence rates of neck and low back pain according to anxiety and depression in Inactive and Active BC Survivors”. What exactly does “according to anxiety and depression” mean?
Response 8: Thanks for the comment. We adjusted this part of the text. Please refer to the page 6 or see below:
“Figure 1. Prevalence of neck and low back pain stratified by anxiety and depression status in physically inactive and active breast cancer survivors”
Response 9: Throughout the manuscript, it should be completely clear when referring to “rates” and when referring to “symptoms” or “intensity”. For example, Figure 1 states that it refers to prevalence rates of neck and low back pain, but the discussion refers to pain intensity. Specifically, on page 7, lines 225-226, it says: “The current study aimed to examine the relationship between anxiety, depression, and pain intensity”. It is suggested that the entire manuscript, including the Discussion, be thoroughly reviewed.
Response 9: Thanks for the comment. We adjusted this part and reviewed all the text. Please refer to the page 7-10.

Round 3
Reviewer 3 Report
Comments and Suggestions for Authors
The revised manuscript "Physical Inactivity Amplifies the Link Between Anxiety, Depression, and Neck Pain in Breast Cancer Survivors" is improved from the second version. However, one of the identified errors remains. Below is a copy of the question raised and the authors' response:
Comments 2: In the Abstract, on lines 28 to 30, it says: “The presence of anxiety, depression, and neck pain was higher in physically inactive women than in active women, however, there was no statistically significant difference.”. But according to the data in Table 1, the score for depression is higher for active women (18.8) than for inactive women (13.7), and the same is true for neck pain, which is higher for active women (37.5) than for inactive women (34.7).
Response 2: Thanks for the comment. Dear Reviewer, the information in the abstract regarding prevalence data refers to Figure 1, which presents a comparison among women with anxiety and depression.
If the sentence in the abstract, which in the corrected manuscript is on lines 25 to 27 and reads “The presence of anxiety, depression, and neck pain was higher in physically inactive women than in active women, however, there was no statistically significant difference.” refers to the data in Figure 1, this text should be phrased correctly with the data from Figure 1. This figure does not present a comparison among women with anxiety and depression, but presents the prevalence of neck and low back pain stratified by anxiety and depression status in physically inactive and active women. Where the results of anxiety and depression for active and inactive women appear is in Table 1, where, as I said in the previous review, the score for depression is higher for active women (18.8) than for inactive women (13.7), and the same is true for neck pain, which is higher for active women (37.5) than for inactive women (34.7), although the differences are not statistically significant.
Author Response
The revised manuscript "Physical Inactivity Amplifies the Link Between Anxiety, Depression, and Neck Pain in Breast Cancer Survivors" is improved from the second version. However, one of the identified errors remains. Below is a copy of the question raised and the authors' response:
“Comments 2: In the Abstract, on lines 28 to 30, it says: “The presence of anxiety, depression, and neck pain was higher in physically inactive women than in active women, however, there was no statistically significant difference.”. But according to the data in Table 1, the score for depression is higher for active women (18.8) than for inactive women (13.7), and the same is true for neck pain, which is higher for active women (37.5) than for inactive women (34.7).”
“Response 2: Thanks for the comment. Dear Reviewer, the information in the abstract regarding prevalence data refers to Figure 1, which presents a comparison among women with anxiety and depression.”
Comments 1: If the sentence in the abstract, which in the corrected manuscript is on lines 25 to 27 and reads “The presence of anxiety, depression, and neck pain was higher in physically inactive women than in active women, however, there was no statistically significant difference.” refers to the data in Figure 1, this text should be phrased correctly with the data from Figure 1. This figure does not present a comparison among women with anxiety and depression, but presents the prevalence of neck and low back pain stratified by anxiety and depression status in physically inactive and active women. Where the results of anxiety and depression for active and inactive women appear is in Table 1, where, as I said in the previous review, the score for depression is higher for active women (18.8) than for inactive women (13.7), and the same is true for neck pain, which is higher for active women (37.5) than for inactive women (34.7), although the differences are not statistically significant.
Response 1: Thank you very much for your comment. We sincerely apologize for the imprecise way in which the sentence was originally written in the abstract. We fully agree with your observation that the results described did not accurately reflect the data presented in Figure 1. We adjusted this part of the text. Please refer to the page 1 or see below:
“…This cross-sectional study was conducted with 128 women breast cancer survivors. The prevalence of neck pain stratified by anxiety and depression status was higher in women physically inactive than in active women, however, there was no statistically significant difference. Symptoms of anxiety and depression were associated with neck pain only in physically inactive women. No associations were observed between anxiety and depression and low back pain regardless of physical activity levels. This study demonstrates significant results for the association between physical activity level, anxiety and depression, and neck pain in women breast cancer survivors. However, the same association was not found for low back pain...”
